# Fecal Microbiota Transplantation Using Donor Stool Obtained from Exercised Mice Suppresses Colonic Tumor Development Induced by Azoxymethane in High-Fat Diet-Induced Obese Mice

**DOI:** 10.3390/microorganisms13051009

**Published:** 2025-04-27

**Authors:** Hiroshi Matsumoto, Tingting Gu, Shogen Yo, Momoyo Sasahira, Shuzo Monden, Takehiro Ninomiya, Motoyasu Osawa, Osamu Handa, Eiji Umegaki, Akiko Shiotani

**Affiliations:** Department of Gastroenterology, Kawasaki Medical School, Kurashiki 701-0192, Japan; gutingting0529@med.kawasaki-m.ac.jp (T.G.); hamitaro0607@gmail.com (S.Y.); momomo0318@gmail.com (M.S.); mondymondy1801@gmail.com (S.M.); takehiro.nino0614@gmail.com (T.N.); o.m.1976-1017@med.kawasaki-m.ac.jp (M.O.); handao@med.kawasaki-m.ac.jp (O.H.); eumegaki@med.kawasaki-m.ac.jp (E.U.); shiotani@med.kawasaki-m.ac.jp (A.S.)

**Keywords:** fecal microbiota transplantation (FMT), colorectal cancer, obesity, exercise, high-fat diet

## Abstract

The gut microbiota plays an important role in the development of colorectal tumors. However, the underlying mechanisms remain unclear. In this study, we examined the effects of fecal microbiota transplantation (FMT) on azoxymethane (AOM)-induced colorectal tumors in obese mice. We divided the study subjects into the following five groups: high-fat diet (HFD), normal diet (ND), ND+exercise (Ex), HFD+FMT from ND-alone donor (HFD+FMT(ND alone)), and HFD+FMT from ND+Ex donor (HFD+FMT(ND+Ex)). The Ex group performed treadmill exercise for 15 weeks. Thereafter, fecal and colonic mucus samples were extracted for microbiome analysis. The deoxyribonucleic acid sample was collected from the feces and colonic mucosa, and V3–V4 amplicon sequencing analysis of the 16S rRNA gene was performed using MiSeq. The number of polyps was significantly lower in the ND (6.0 ± 1.6) and ND+Ex (1.8 ± 1.3) groups than in the HFD group (11.4 ± 1.5). The ND+Ex group had significantly fewer polyps than the ND group. The HFD+FMT(ND alone) (5.2 ± 0.8) and HFD+FMT(ND+Ex) (2.8 ± 2.6) groups also had significantly fewer polyps than the HFD group. The IL-15 mRNA levels in the colonic tissues were significantly higher in the HFD+FMT(ND alone) group than in the ND group. Fecal ω-muricholic acid concentrations were significantly higher in the HFD+FMT(ND alone) group than in the ND group and in the HFD+FMT(ND+Ex) group than in the ND+Ex group. The ND, ND+Ex, HFD+FMT(ND alone), and HFD+FMT(ND+Ex) groups had a significantly higher abundance of *Lacyobacillaceae* than the HFD group. In the FMT group, *Erysipelotrichaceae* and *Tannerellaceae* were significantly less abundant. Compared with the HFD group, the ND, ND+Ex, HFD+FMT(ND alone), and HFD+FMT(ND+Ex) groups had a significantly higher abundance of *Muribaculaceae* and a significantly higher abundance of *Lactobacillaceae* and *Rikenellaceae* in common among the ND and ND+Ex groups. The common and significantly less common species were *Bacteroidaceae* in the FMT group and *Lactobacillaceae* and *Rikenellaceae* in the ND alone and ND+Ex groups. *Bacteroidaceae* and *Lachnospiraceae* were significantly less common in the FMT group. We found that FMT inhibited AOM-induced colorectal tumorigenesis in obese mice. Furthermore, the fecal concentrations of short-chain fatty acids, bile acids, microbiota, and mucosa-associated microbiota differed between the FMT and diet/EX groups, suggesting that the inhibitory effect of FMT on colorectal tumorigenesis may be due to mechanisms different from those of ND alone and ND+Ex.

## 1. Introduction

Colorectal cancer (CRC) is the third most commonly diagnosed cancer type and the most deadly type of cancer worldwide [1]. As one of the most important factors, excessive dietary fat intake is strongly associated with an increased CRC risk [2]. However, the underlying mechanism of CRC development is still largely unclear. Different from the other cancer types, CRC directly interacts with trillions of gut microorganisms during tumor development. The composition of the gut microbiota is influenced by multiple factors, including diet, medication, and genetic alteration, and an altered microbial profile can induce dysbiosis and CRC [3]. Particularly, the gut microbiota is shown to be perturbed at a very early stage of colorectal cancer formation [4] and becomes aggravated during disease progression [5]. Furthermore, microbiome-derived metabolites, including bile and short-chain fatty acids (SFAs), may also contribute to or prevent the development of CRC [6]. Growing evidence supports that the characteristics of the gut microbiota can be tumorigenic or protective against CRC. CRC is characterized by considerable dysbiosis, including reduced SFA-producing bacteria and the changes seen with CRC.

In the development of colorectal adenomas and cancers, the involvement of the intestinal microbiota and its metabolites has recently been reported [7,8]. The findings suggest that certain microbial species promote tumorigenesis. Particularly, microorganisms such as *Fusobacterium nucleatum*, *Streptococcus bovis*/*gallolyticus*, *Escherichia coli*, and *Bacteroides fragilis* are abundant in patients with colorectal adenocarcinoma. These microorganisms may promote the development of CRC through their ability to adhere to colonic cells, repress tumor suppressor genes, activate oncogenes, and modulate genotoxicity. The intestinal microbiota is present not only in the fecal lumen but also in the mucus of the gastrointestinal tract, which is then referred to as the mucosa-associated microbiota (MAM). Interestingly, the microbiota present in the feces and mucus differ [9]. The MAM is in contact with the intestinal epithelium and may have a greater influence on colorectal tumors than on the luminal microbiota [5,8]. As for the MAM of colorectal tumors, Pseudomonas, Helicobacter, and Acinetobacter, which are classified in the genus Proteobacteria, have been reported to be increased, whereas Bacteroides are decreased [10].

Fecal microbiota transplantation (FMT) is a method used to directly change the recipient’s gut microbiota to normalize its composition and gain a therapeutic benefit [11]. The history of FMT can be traced back to the 4th century and has been highly regarded since 2013, when the United States Food and Drug Administration approved the use of FMT for treating recurrent and refractory *Clostridiodes (Clostridium) difficile* (CD) infection.

Currently, the primary indication for FMT is recurrent CD infection. Clinically, FMT has been reported for inflammatory bowel disease and irritable bowel syndrome. FMT is being explored as a potential therapeutic strategy for cancer, including colon tumors. This approach involves reconstructing the intestinal microbiota, which can influence cancer development and progression. It works by modulating the relevant factors, such as bile acid metabolism and the efficacy of immunotherapy. The gut microbiota, when altered (a condition known as dysbiosis), can affect tumorigenesis by activating certain pathways, inducing inflammation, and causing damage to the host’s deoxyribonucleic acid (DNA). The goal of FMT in cancer management is to restore a healthy balance of the gut microbiota, thereby potentially reducing the body’s cancer-promoting conditions. However, the FMT’s effect on tumors is unknown.

Effective FMT depends considerably on the quality and characteristics of the donor stool. The screening process for selecting suitable donors is quite rigorous and involves several important factors, including microbial composition, the stability of the gut microbiota, age and body mass index criteria, and infectious agents.

The donor stool is evaluated for the presence of beneficial and harmful microbial taxa, as well as overall microbial richness. This includes assessing for specific pathogens and ensuring a high richness of beneficial bacteria. The stability of the donor’s gut microbiota over time is also considered and is important for ensuring consistency in FMT treatments. This comprehensive approach to donor screening ensures that the stool used in FMT is safe, effective, and of high quality, catering to the specific needs of the FMT recipients. No study has examined what donor stools are effective for colorectal tumors.

Previously, we reported the inhibitory effect of exercise on AOM-induced colorectal tumors in high-fat diet-induced obese mice, focusing on the intestinal microbiota, especially the mucosa-associated microflora. In the present study, we aimed to investigate the inhibitory effect of FMT on AOM-induced colorectal tumors in high-fat diet-induced obese mice. We found that FMT may suppress the tumors through a mechanism other than the efficacy of the dietary modification.

## 2. Materials and Methods

The study protocol was conducted in accordance with the Kawasaki Animal Regulations (approval no. 22-098).

### 2.1. Animals

Balb/c female mice were purchased at 4 weeks, with six to eight animals per group. A 12 h light/dark cycle was maintained, with a room temperature of 22 ± 1 °C, humidity of 55–60%, and ad libitum access to chow and sterilized water under specific-pathogen-free (SPF) conditions.

### 2.2. AOM-Induced Colorectal Cancer Model

Colorectal cancers were used in the AOM-induced model [12,13,14]. AOM was administered intraperitoneally at 10 mg/kg body weight weekly for a total of six administrations (6–11 weeks). Colon cancer development was observed every 4 weeks via endoscopy (AVS endoscopy system, OLYMPUS, Tokyo, Japan). Finally, the colon was dissected during autopsy at 26 weeks. We measured the colon polyp size using a ruler at a 5× magnification. Histopathological analysis of the colon polyps in the mice was also performed [15].

### 2.3. Study Design

The mice were divided into the following five groups (*n* = 6–8): HFD (high-fat diet, *n* = 8); ND (normal-fat diet, *n* = 8); ND with exercise (Ex) (ND+Ex, *n* = 8); HFD with FMT donor ND (HFD+FMT(ND alone), *n* = 6); and HFD with FMT donor ND with exercise (Ex) (HFD+FMT(ND+Ex), *n* = 6) (Figure 1).

The diet was changed after acclimation for 1 week. The diets were an ND and an HFD (HFD 60^®^; total calories: 5062 kcal/kg; calorie ratio (%): protein (18.2), fat (62.2), and digestible carbohydrates (19.6)). The Ex group included treadmill exercise [16,17]. The running speed was 18 m/min for 30 min/day for 5 days a week. The Ex group was started at 6 weeks of age for 2 weeks and continued for 20 weeks from 8 to 26 weeks.

### 2.4. Fecal Sample Collection and FMT

For comparative purposes, the feces of the mice were collected twice for 16S ribosomal RNA (rRNA) gene amplicon sequencing, once at the beginning of week 12 (i.e., before FMT) and once at the end of week 24 (i.e., after FMT). An amount of 100 mg of feces from each mouse was stored in 1 mL of Inhibit EX Buffer (Qiagen, Gaithersburg, MD, USA) at −80 °C for ≤1 week until DNA extraction. After 4 weeks of a formal exercise regimen (i.e., at week 12), feces from the exercised mice were collected fresh for daily FMT [16,18], which was conducted under sterile conditions under a laminar flow hood. The feces from the exercised mice were pooled by cage, and 100 mg (approximately 5–6 fecal pellets) was re-suspended in 1 mL of sterile saline. The solution was vigorously mixed for 10 s before centrifugation at 800× *g* for 3 min. The supernatant (approximately 500–600 μL) was collected and administered by oral gavage within 10 min to minimize the changes in the microbial contents [19].

According to the established microbial depletion and recolonization protocol [20,21,22,23], a combination of ciprofloxacin (0.2 g/L) and metronidazole (1 g/L) was added to the drinking water of the FMT recipients for 2 days (i.e., the weekend of week 11) before FMT to ensure that the outcomes were compatible with the clinical guidelines for FMT in humans [24]. Ciprofloxacin plus metronidazole was chosen because it is one of the first-line antibiotic regimens recommended for treating abdominal infections in adults [25]. Non-FMT recipients did not receive any antibiotics. The antibiotics used were obtained from Sigma-Aldrich Corp. (St. Louis, MO, USA). From week 12, FMT was conducted each weekday, with each recipient administered 100 μL of fecal supernatant by oral gavage until week 26.

### 2.5. Colonic Mucosa Sample Collection

At the end of the experiments, the mice were euthanized by deep anesthesia using carbon dioxide inhalation. The abdominal cavities were opened within 10 min, and the intestinal segments were removed. Then, the colon was cut longitudinally and separately, and the intestinal luminal contents were stored in sterile EP tubes. After removing the contents, the remaining colon was rinsed in precooled sterile phosphate buffer (PBS) twice to separate it from the mucus, loosely bound bacteria, and digestive substances adhering to the intestinal wall. Then, the intestinal mucosa was scraped from the inner intestinal surface using a cover glass, collected, and stored in sterile EP tubes [26].

### 2.6. Metabolic Marker Profiles of Mice

For each mouse, the body weight was measured weekly, and after euthanizing the mouse after an overnight fast at the end of the experimental period, the chest was opened, and blood was drawn from the left ventricular apex of the heart. Serum samples were obtained by the Animal Center within 1 week for the measurement of fasting blood glucose and serum cholesterol levels using an analyzer.

### 2.7. RT Quantitative PCR (qPCR)

RT-qPCR tests were performed using the StepOnePlus™ Real-Time PCR Systems (Thermo Fisher Scientific, Waltham, MA, USA) with a PowerUp™ SYBR™ Green Master Mix (Thermo Fisher Scientific). The primers used for the RT-qPCR experiments are provided in Table 1. The mouse actin-beta (Actb) expression was evaluated as an internal control. All reactions were performed three times. The PCR conditions were as follows: after initial denaturing at 95 °C for 2 min, 40 cycles were performed at 95 °C for 15 s and at 60 °C for 1 min, followed by a melting-curve analysis (95 °C for 15 s, 60 °C for 1 min, and 95 °C for 15 s).

### 2.8. Gut Microbiome Analysis

The fecal and colonic mucus samples were analyzed. The fecal samples were collected at 26 weeks [100 mg of feces/mouse in 1 mL of Inhibit EX Buffer (approximately 5–6 feces) from each mouse and stored at a temperature of −80 °C in 1 mL of EX Buffer (Qiagen)]. The colonic mucus samples were collected from each mouse and stored at a temperature of −80 °C. To obtain the colonic mucus samples, the colon was incised in the long axis and rinsed lightly with PBS, and the lumen was scraped with a glass slide to collect the mucus [26,27]. Bacterial DNA was extracted from the collected samples using the bead-crushing method (QIAamp PowerFecal, Tokyo, Japan, QIAGEN), and QIIME was used to perform the V3–V4 amplicon sequencing analysis of 16S rRNA genes at the genus level, identify microorganisms at the genus level, and investigate the bacterial composition and diversity. This advanced genomic study was conducted at the Department of Bacteriology, Kyushu University. The taxonomic and functional profiles were further analyzed in the STAMP software v2.1.3

### 2.9. Fecal Metabolome Analysis

The fecal samples collected at 26 weeks were analyzed for SFs and non-conjugated bile acids (pH-buffered post-column conductivity detection method; TechnoSurga).

### 2.10. Bioinformatics Analysis

Sequence data processing, consisting of quality filtering, chimera checking, operational taxonomic unit (OTU) definition, and taxonomic assignment, was accomplished using the combination of QIIME 1.9.0 (https://qiime2.org/ (accessed on 26 April 2025)), USEARCH 9.2.4 (https://cryptick-lab.github.io/NGS-Analysis/_site/usearch-previous.html (accessed on 26 April 2025)), UCHIME 4.2.40 (https://www.drive5.com/usearch/ (accessed on 26 April 2025)), and VSEARCH 2.4.3 (https://github.com/torognes/vsearch/releases/tag/v2.28.1 (accessed on 26 April 2025)). Singletons were eliminated, and 97% sequence similarity OTUs were taxonomically assigned using the RDP classifier v2.10.2 with the Greengenes database version 13.8.

### 2.11. Statistical Analysis

The differences in families among the lines were analyzed using the Statistical Analysis of Metagenomic Profiles (STAMP) software (version 2.1.3). The diversity and relative abundance of bacterial genera indices were compared between three groups by Kruskal–Wallis analysis and between two groups by the Mann–Whitney U test. The categorical data were analyzed using Fisher’s exact test. Statistical analyses were performed using SPSS (version 25 for Windows, IBM Japan, Ltd., Tokyo, Japan). Statistical significance was set at a *p*-value of <0.05.

## 3. Results

### 3.1. Mouse Body Weight Change by Diet and FMT

The ND and ND+Ex groups showed a greater weight loss effect than the HFD group, but not both the FMT groups, regardless of the donor’s stool (Figure 2).

### 3.2. FMT’s Effect Against AOM-Induced Colorectal Tumor Count

The number of polyps was significantly lower in the ND (6.0 ± 1.6) and ND+Ex (1.8 ± 1.3) groups than in the HFD group (11.4 ± 1.5). The ND+Ex group had significantly fewer polyps than the ND group, and the FMT(ND) (5.2 ± 0.8), and FMT(ND+Ex) (2.8 ± 2.6) groups also had significantly fewer polyps than the HFD group. (Figure 3A,B). Regarding the polyp size, the other four groups had fewer polyps larger than 3 mm compared with the HFD group (Figure 3C). The colorectal tumors were analyzed macroscopically and histopathologically (Appendix A). Microscopically, these tumors displayed the characteristics of intraepithelial neoplasia, with changes ranging from low-grade dysplasia to high-grade dysplasia/intramucosal carcinoma.

### 3.3. Blood Glucose (BS) and Total Cholesterol Levels

The ND (105.8 ± 36.7), ND+Ex (78.3 ± 13.3), HFD+FMT(ND) (116.0 ± 42.1), and HFD+FMT(ND+Ex) groups (117.5 ± 45.0) had significantly lower blood glucose levels than the HFD group (241.7 ± 74.6) (Figure 4A,B). The ND and ND+Ex groups showed significantly lower total cholesterol levels than the HFD group, but the HFD+FMT(ND) and HFD+FMT(ND+Ex) groups did not.

### 3.4. SFA Analysis of Fecal Samples

The succinic, acetic, propionic, and butyric acid concentrations in the feces were significantly higher in the ND and ND+Ex groups than in the HFD group (Figure 5A–E). Contrarily, there was no significant difference in these concentrations between the HFD+FMT(ND alone), HFD+FMT(ND+Ex), and HFD groups. Meanwhile, there were significant differences in succinic, acetic, propionic, and butyric acid levels for ND versus FMT mice.

### 3.5. Nonconjugated Bile Acid Analysis of Feces

Fecal ω-muricholic acid concentrations were significantly higher in the HFD+FMT(ND) group than in the ND group and in the HFD+FMT(ND+Ex) group than in the ND+Ex group (Figure 6A–F).

### 3.6. Cytokine and Myokine Expressions in Colonic Tumors

The IL-15 levels in the colonic tissues were significantly higher in the HFD+FMT(ND) group than in the ND group. The irisin level in the colonic tissue was significantly lower in the HFD+FMT(ND) and HFD+FMT(ND) groups than in the HFD group (Figure 7A–F).

### 3.7. Changes in Fecal Microbiota

Table 2 shows the strains with significant differences between the five groups (Table 2). The ND, ND+Ex, HFD+FMT(ND alone), and HFD+FMT(ND+Ex) groups showed a significantly higher abundance of Lacyobacillaceae than the HFD group.

In the taxonomic analysis (Figure 8A) comparing the five groups, FMT resulted in more Verrucomicrobiota than the other three groups and less Bacteroidota.

The comparisons between two groups in Figure 8B–E show the list of bacterial species that showed significant differences between the four groups. 

### 3.8. Changes in MAM

Table 3 shows the strains with significant differences between the five groups. The ND, ND+Ex, HFD+FMT(ND alone), and HFD+FMT(ND+Ex) groups had a significantly higher abundance of Muribaculaceae than the HFD group and significantly more Lactobacillaceae and Rikenellaceae in common among the ND and ND+Ex groups. The common and significantly less common species in the FMT group were Bacteroidaceae, and the common and significantly less common species in the ND and ND+Ex groups were Lactobacillaceae and Rikenellaceae. Bacteroidaceae and Lachnospiraceae were significantly less common in the FMT group (Table 3).

In the taxonomic analysis, the ND and ND+Ex groups had more Bacteroidota than the HFD group, and the FMT group had more Verrucomicrobiota than the other groups.

The comparisons between two groups are displayed in Figure 9B–E. (B–E) show the comparison of four groups, displaying the list of bacterial species that showed significant differences between the four groups. 

## 4. Discussion

In the present study, we found that FMT inhibited the AOM-induced colorectal tumorigenesis in obese mice. Furthermore, the fecal concentrations of SFAs, bile acids, microbiota, and MAM were different between the FMT and diet/Ex groups, suggesting that the inhibitory effect of FMT on colorectal tumorigenesis may be due to mechanisms different from those of the ND and ND+Ex groups.

The study of the role of the gut microbiota in CRC is an emerging field of research. The regulation of the gut microbiota with the aim of reversing established microbial dysbiosis is an emerging strategy for preventing and treating CRC. Various strategies have been employed, including the use of probiotics, prebiotics, postbiotics, antibiotics, and FMT. These strategies have shown mechanistically promising results in correcting the microbiota composition, modulating the innate immune system, enhancing the intestinal barrier function, preventing pathogenic colony formation, and exerting selective cytotoxicity against tumor cells. In the present study, we showed for the first time that fecal transplantation is as effective as dietary modification in preventing colorectal tumorigenesis in diet-induced obese mice. Particularly, we demonstrated for the first time that transplantation of a normal diet suppresses colorectal cancers caused by a high-fat diet. Furthermore, we found that the effect of fecal transplantation was different from that of dietary modification. To date, five reports have examined the effect of FMT using mouse colorectal cancer models [28,29,30,31,32]. Four studies have also investigated the effect of FMT on an AOM-induced colorectal cancer model. As donor feces, there are two reports of using normal feces, CRC feces, and transplanted feces from ulcerative colitis patients. This is the first report on the effect of FMT on AOM-induced colorectal cancers in diet-induced obese mice.

In the present study, FMT considerably increased the IL-15 expression in the colonic cancer area. This is likely related to the colorectal cancer-suppressive effect of FMT. IL-15 is a cytokine that induces the activation of NK and T cells in the intestinal mucosa and exhibits tumor-suppressive effects [33]. Interestingly, IL-15 has been reported to suppress intestinal inflammation and even exhibit tumor-suppressive effects, regardless of the SFAs [34,35]. In the present study, colonic suppression by FMT also increased the IL-15 expression without elevating the SFAs in the feces. IL-15 could be used as an immunostimulant in cancer therapy [36]. IL-15 has been implicated in chronic diseases and may be involved in the antitumor effects of colon carcinogenesis. IL-15 generates and maintains long-term CD8+ T-cell immunity against *Toxoplasma gondii* and controls intracellular pathogen infection [37]. IL-15 deficiency in mice contributes to the downregulation of IFNγ-producing CD4+ T-cell responses to acute *T. gondii* infection. IL-15 induces hyperplasia, which may have multiple functions. The limited role of IL-15 in NK and CD8+ T-cell development suggests that other cytokines may exist to compensate for the deficiency of the IL-15 gene. In the present study, histological studies were not performed; thus, we were unable to examine which cells produce IL-15, NK cell expression, and so on. Future studies on the immune response in the intestinal mucosal tissues are needed.

In the present study, we confirmed that fecal transplantation increases the number of *Lactobacillaceae*. *Lactobacillaceae delays* tumor development by increasing the NK and CD8+ T-cell infiltration and increasing IFNγ production in the cancer microenvironment [38]. We observed an increase in IL-15 expression as described above, and IL-15 had an inhibitory effect on tumor development via the NK cells. Therefore, it is possible that the mechanism of suppressing colorectal cancers by FMT in this model was tumor suppression by increased IL-15 expression due to an increased abundance of *Lactobacillaceae*. Additionally, research on the effects of *Lactobacillaceae* probiotics on the colonic microbiota and metabolite production in patients with cystic fibrosis indicated beneficial shifts in the composition of cystic fibrosis. These changes included an increase in the number of beneficial bacteria and a reduction in the pathogenic genera, which might contribute to an overall environment that is less conducive to tumor development [39]. To date, there are reports of an increase in *Lactobacillaceae* caused by FMT [40], but it is still unclear why *Lactobacillaceae* is increased by FMT, requiring further investigation.

In the present study, we found that fecal transplantation increased the concentration of muricholic acid in the feces. The association between CRC and bile acids has been reported previously, and muricholic acid has been reported to act in a manner that counteracts the injurious properties of DCA in colorectal tissues; moreover, muricholic acid has a CRC-preventive effect [41]. Taurated muricholic acid has also been reported to reduce FXR activity and improve insulin resistance [42]. We found that FMT enhanced the gut’s bacterial bile acid metabolism and delayed the development of impaired glucose tolerance relative to the placebo control group [43]. The FMT-enriched bacteria involved in intestinal bile acid metabolism include *Desulfovibrio fairfieldensis* and *Clostridium hylemonae*. To identify the candidate bacteria involved in intestinal bile acid metabolism, we focused on the bile acid products of intestinal bacterial metabolism and evaluated the correlation between bacterial species abundance and bile acid profiles. *Bacteroides ovatus* and *Phocaeicola dorei* showed a positive correlation with unconjugated bile acids. *Bifidobacterium adolescentis*, *Collinsella aerofaciens*, and *Faecalibacterium prausnitzii* were positively correlated with secondary bile acids. However, these bacteria were not detected in this study. Previous studies were conducted in humans. Further studies are needed on the effect of FMT on mouse bile acid metabolism, as it differs from human bile acid metabolism.

In the present study, we also showed for the first time that the MAM in a mouse colon tumor model subjected to FMT showed an increase in *Bacteroidaceae* and *Muribaculoceae* and a decrease in *Lachnospiraceae*. However, the changes in the abundances of *Muribaculaceae* and *Lachnospiraceae* were also observed in the ND alone and ND+EX groups. Thus, FMT did not result in any characteristic changes in the MAM that differed from the fecal matter. Recently, Manasvini et al. reported on the differences between the crypt-associated microbiota (CAM) and MAM by FMT [44]. They showed that the CAM is altered by FMT. We characterized the CAM in patients with ulcerative colitis (UC) before and after FMT with an anti-inflammatory diet (FMT-AID). The differences in the composition of the CAM and its interaction with the MAM were compared between the non-IBD controls and UC patients (*n* = 26) before and after FMT. Unlike the CAM, the MAM was dominated by aerobic Actinobacteria and Proteobacteria members, indicating diversity and resilience. The present study did not examine the differences between the MAM and CAM; however, this should be examined in the future to determine how the MAM and CAM in this model are altered by FMT.

Several factors influence the success of FMT, including those related to the donor and recipient (the diversity and specific composition of the gut microbiota, immune system, host genetics, and so on), as well as those related to the working protocol (volume and frequency of fecal infusion, route of administration, and adjuvant therapy) [45,46].

In the present study, the tumor-suppressive effect of FMT did not differ much by the donor feces. The advantage of transplanting mouse feces in mice with ND and exercise was not clear. Regarding the tumor-suppressive effect of exercise, a mechanism due to increased fecal SFAs and an anti-inflammatory effect, strengthening the intestinal barrier function, has been considered [47]. In the present study, there was no increase in SFAs in the feces of the FMT group, suggesting that the effect of SFAs on tumors is unlikely. The FMT protocol used in this study followed a previous protocol of FMT, which was effective against obesity and its metabolic abnormalities [16]. Oral administration five times a week for 14 weeks was performed. However, additional studies are needed to determine if FMT in this model is successful, as the number of doses and route of administration differed between studies.

The levels of succinic acid, propionic acid, and butyric acid in the feces were significantly lower in the HFD group than in the ND group. Furthermore, they were also significantly lower in the FMT group than in the ND group. Even when the donor feces for FMT were used as normal feces, the levels of SFAs were still low. Propionic acid is abundantly synthesized by Bacteroidetes and Negativicutes, utilizing succinic acid [48]. In this study, the FMT group had fewer Bacteroidota than the other three groups, and this may be one of the reasons for the lower propionic acid levels.

This study had several limitations. First, the number of mice in each group was small. Second, because histopathological studies were not performed, we were unable to identify the mechanism of the increased IL-15 production and the involved cells. Third, we were unable to show the changes in the MAM over time with FMT or examine how the FMT-induced changes in the MAM contributed to the suppression of tumorigenesis. The use of antibiotics prior to FMT in non-infectious diseases is more problematic because they may exacerbate the microbiome changes [49]. However, the administration of antibiotics prior to FMT reportedly increases the likelihood that the donor microbiome will be viable in mice [50]. Finally, microbiota-based therapy for CRC has advantages and limitations [28]. Various dietary interventions, including the use of prebiotics and probiotics and FMT, have been used to modulate the gut microbiota for the prevention and treatment of CRC; however, further investigation is still needed to determine which approaches are most effective.

## 5. Conclusions

We found that FMT inhibited AOM-induced colorectal tumorigenesis in obese mice. Furthermore, the fecal concentrations of short-chain fatty acids, bile acids, microbiota, and mucosa-associated microbiota differed between the FMT and diet/EX groups, suggesting that the inhibitory effect of FMT on colorectal tumorigenesis may be due to mechanisms different from those of the ND alone and ND+Ex groups.

## Figures and Tables

**Figure 1 microorganisms-13-01009-f001:**
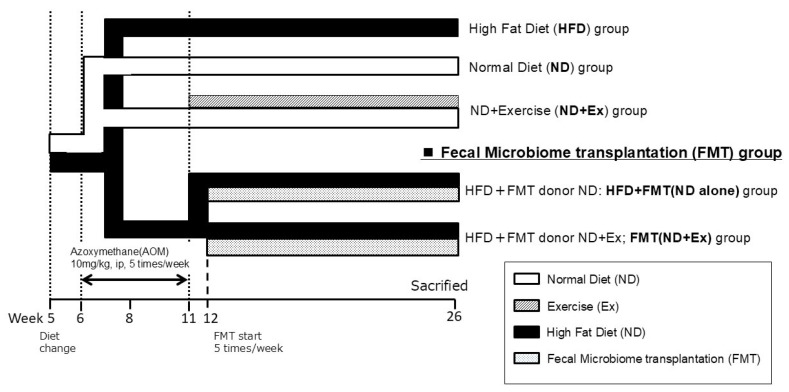
Study design. Five groups: ‘high-fat’ diet (HFD), normal diet (ND), ND+exercise (Ex), HFD+ fecal microbiome transplantation (FMT) using ND mice feces (ND alone), and HFD+FMT using ND+Ex feces (FMT(ND+Ex)). ip, intraperitoneal injection; AOM, azoxymethane.

**Figure 2 microorganisms-13-01009-f002:**
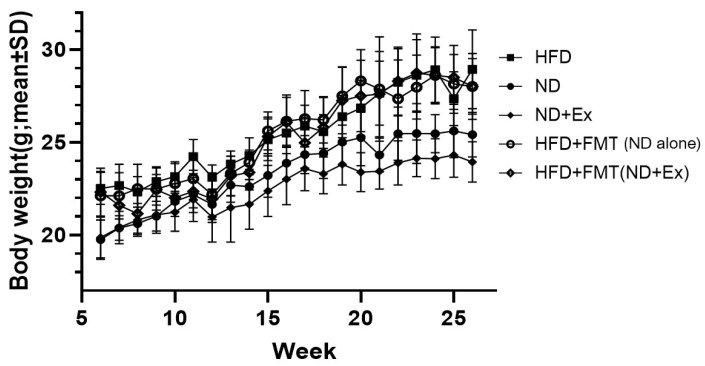
Mouse body weight change. At 26 weeks, the highest weight was in the high-fat diet (HFD) group and the lowest in the normal diet (ND) plus exercise (ND+Ex) group. There was no significant difference between HFD, HFD+FMT (ND alone), and HFD+FMT (ND+Ex).

**Figure 3 microorganisms-13-01009-f003:**
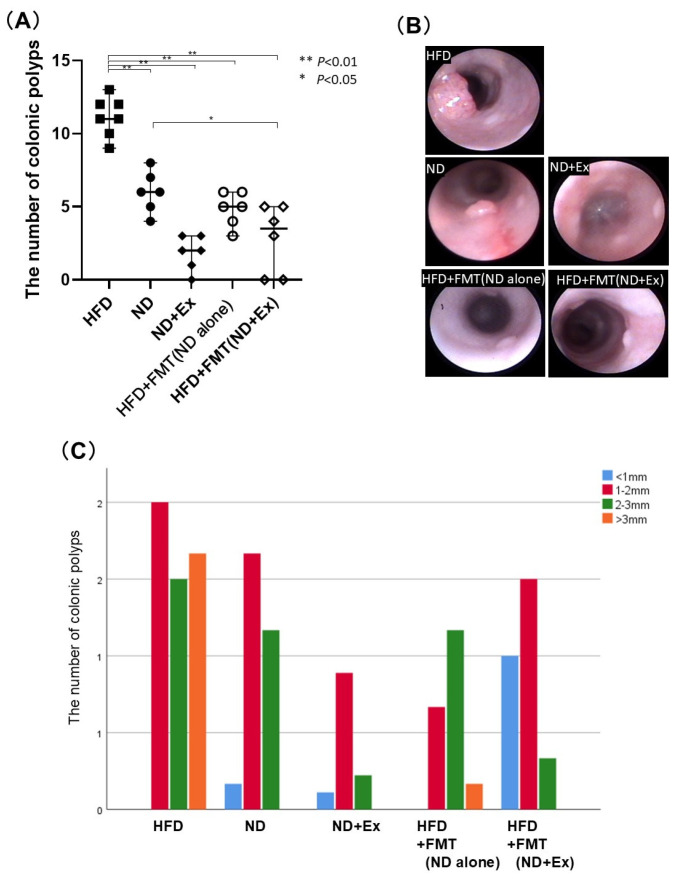
(**A**) Comparison of number of total colonic polyps; (**B**) endoscopic findings; (**C**) comparison of number of colonic polyps and polyp size distribution: The HFD group had the highest number of polyps and the largest polyps. The ND+Ex group had the fewest polyps and the smallest polyp size.

**Figure 4 microorganisms-13-01009-f004:**
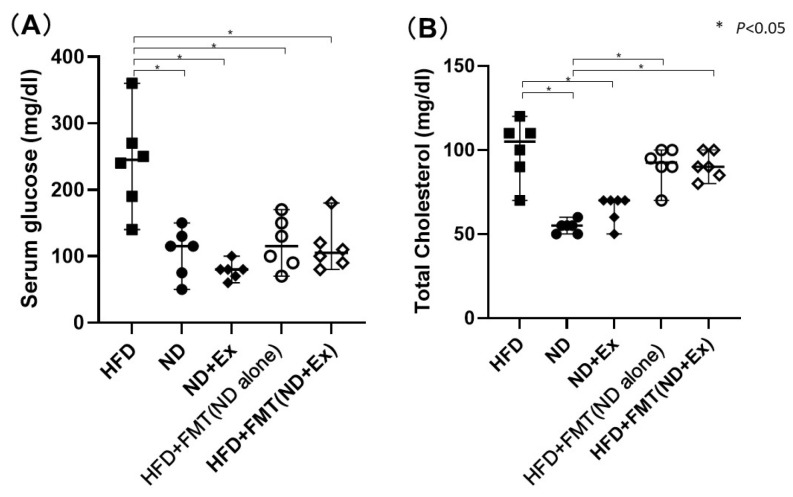
(**A**) Blood glucose and (**B**) total cholesterol levels: serum blood glucose levels were higher in the HFD group than in the others and decreased with the addition of FMT.

**Figure 5 microorganisms-13-01009-f005:**
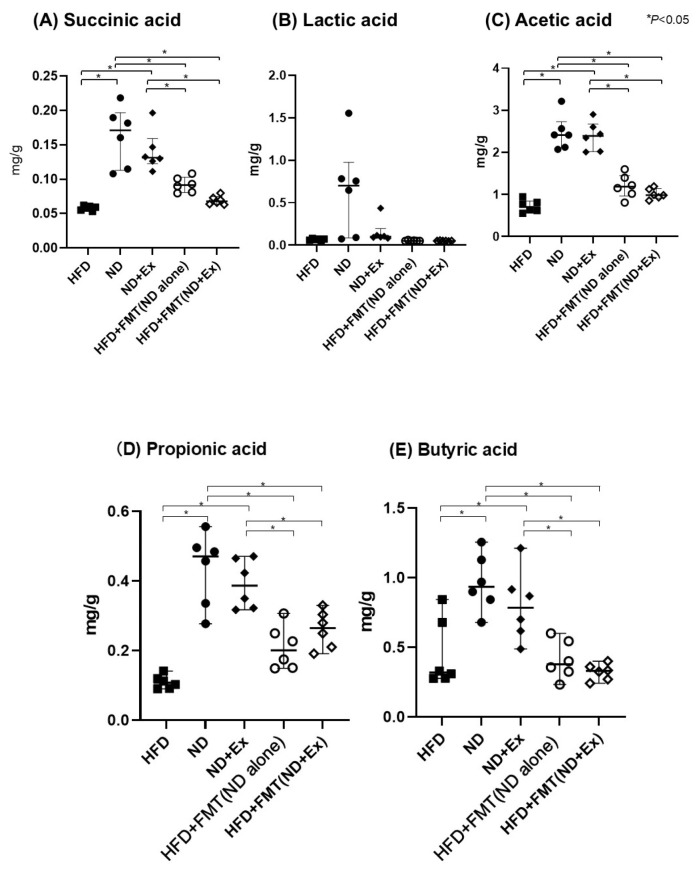
**Fecal SFA levels:** (**A**) succinic acid, (**B**) lactic acid, (**C**) acetic acid, (**D**) propionic acid, and (**E**) butyric acid. Succinic acid and acetic acid were significantly lower in the ND and ND+Ex groups than in the HFD group. Succinic acid and acetic acid levels were significantly higher in the ND+Ex group than in the HFD group.

**Figure 6 microorganisms-13-01009-f006:**
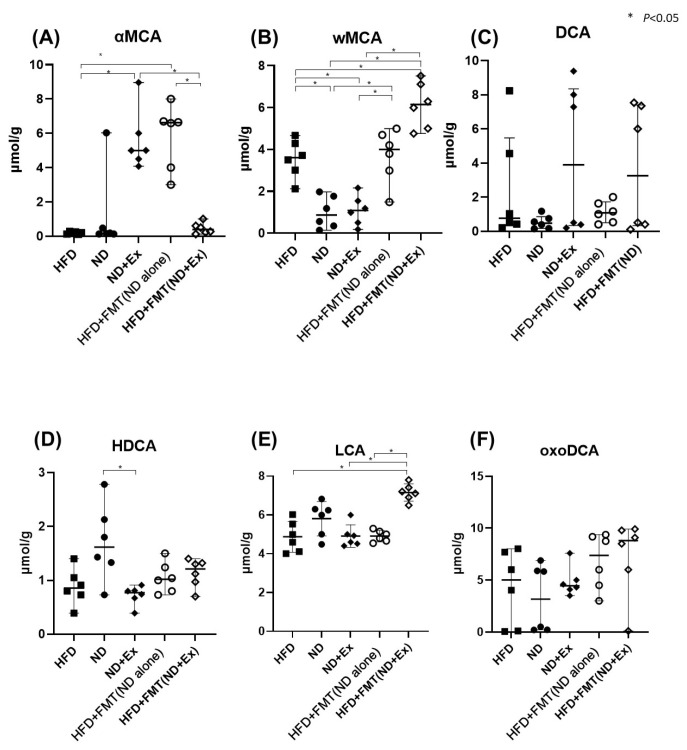
Nonconjugated bile acid level in feces. (**A**) Alpha-MCA, (**B**) omage MCA, (**C**) DCA, (**D**) HDCA, (**E**) LCA, and (**F**) oxoDCA. The HFD group had significantly lower TNFα and IL-6 levels in colon tissue than the ND group. Exercise did not affect cytokines, and SPARC was significantly higher only in the ND group.

**Figure 7 microorganisms-13-01009-f007:**
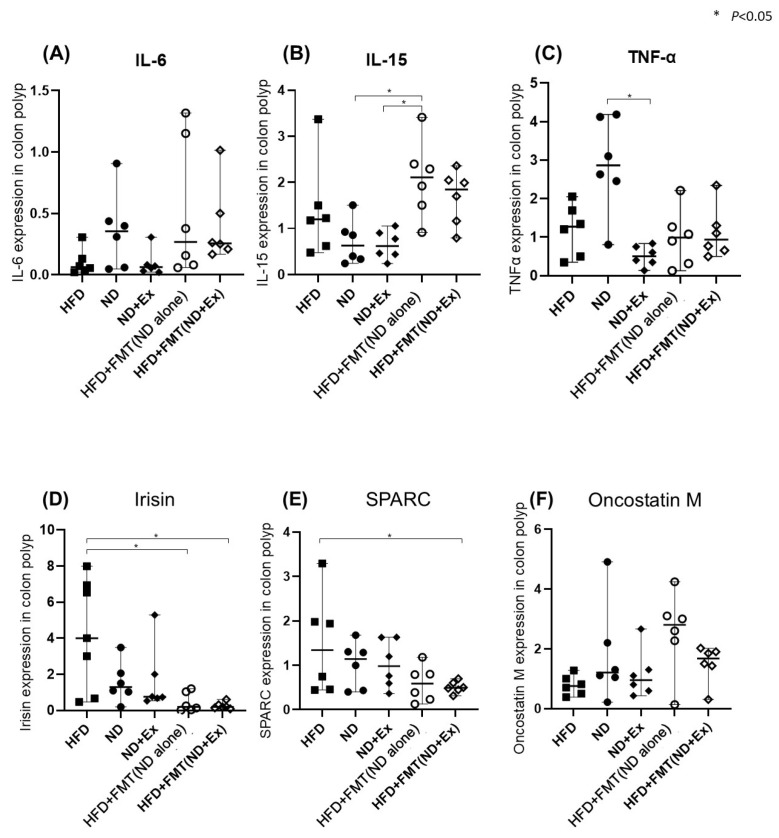
Cytokine and myokine expression by qPCR in colonic tumors. (**A**) IL-6, (**B**) TNF-alpha, (**C**) IL-15, (**D**) SPARC, (**E**) oncostatin M, and (**F**) irisin. The HFD group had significantly lower TNFα and IL-6 levels in colon tissue than the ND group. Exercise did not affect cytokines, and SPARC was significantly higher only in the ND group.

**Figure 8 microorganisms-13-01009-f008:**
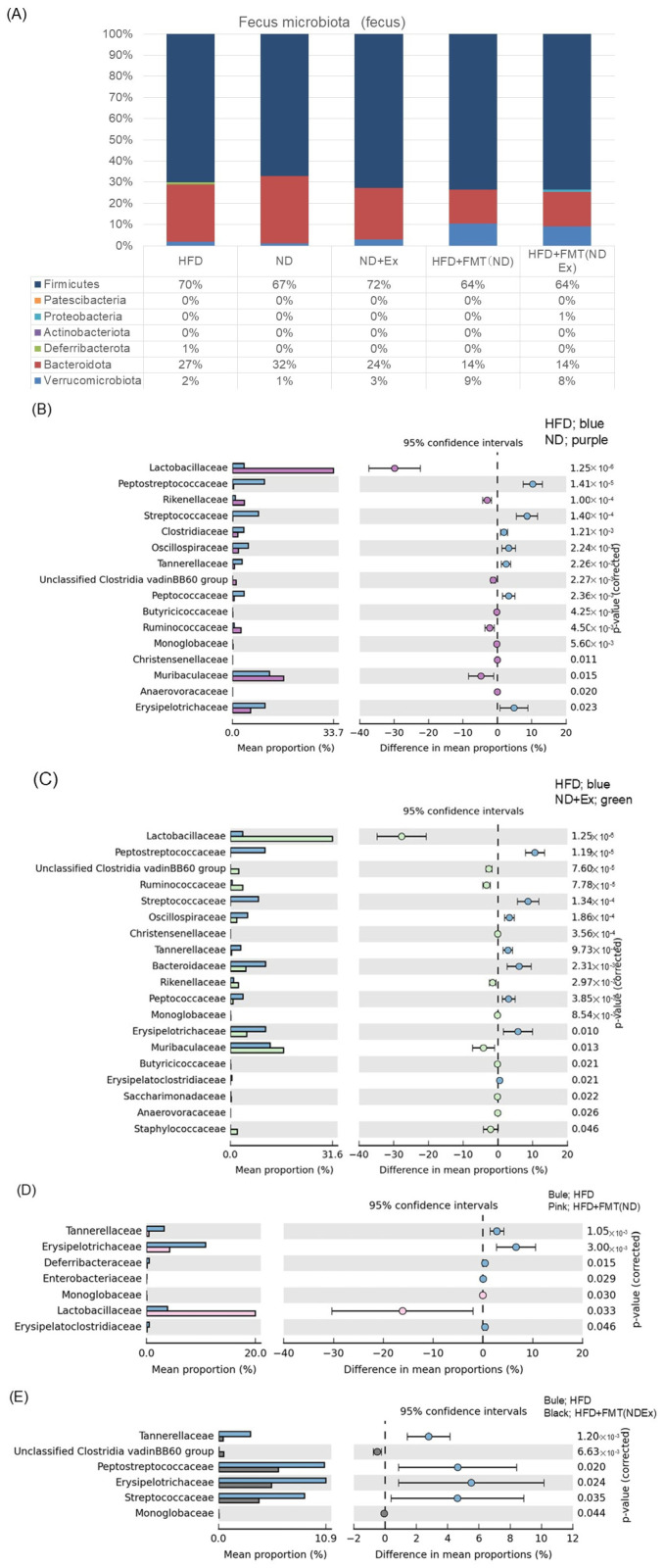
Fecal microbiota analysis. (**A**) Taxonomic analysis. (**B**,**C**) Comparison of 4 groups: list of bacterial species that showed significant differences between the four groups. (**D**) Bacterial species that showed significant differences between the two groups of ND and ND+Ex. (**E**) Bacterial species that showed significant differences between the two groups of HFD and HFD+Ex. In the FMT group, Erysipelotrichaceae and Tannerellaceae were significantly less abundant.

**Figure 9 microorganisms-13-01009-f009:**
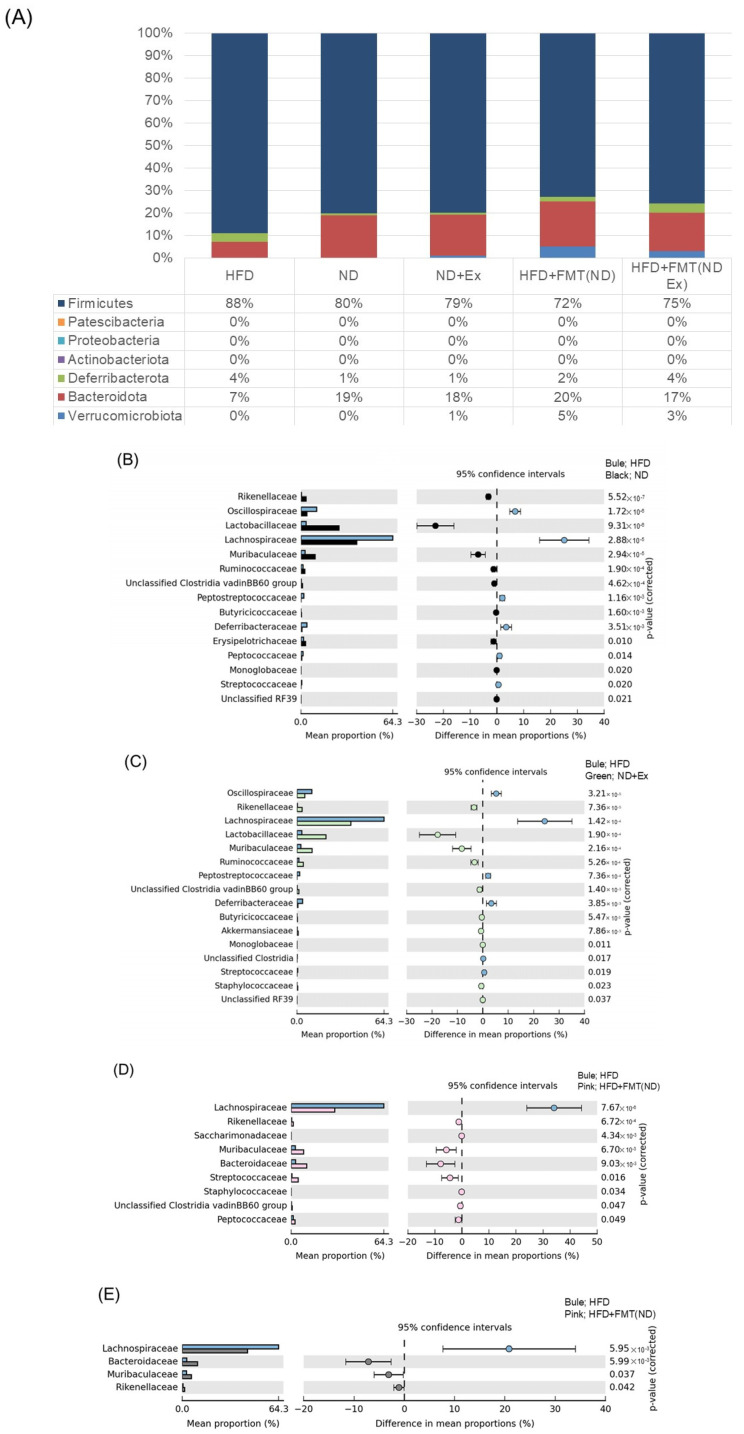
Mucosa-associated microbiota (MAM) analysis. (**A**) Taxonomic analysis. (**B**–**E**) Comparison of 4 groups: list of bacterial species that showed significant differences between the four groups.

**Table 1 microorganisms-13-01009-t001:** PCR primer sequences.

Target Gene		Sequence	Length
IL-6	FP	CGGCCTTCCCTACTTCACAAGTCCG	66
	RP	CAGGTCTGTTGGGAGTGGTATCC	
TNF-alpha	FP	CAACCATCAAGGACTCAAATGG	74
	RP	CCTTTGCAGAACTCAGGAATGGACATTCG	
IL-15	FP	CATCCATCTCGTGCTACTTGTG	112
	RP	GCCTCTGTTTTAGGGAGACCT	
SPARC	FP	CCACACGTTTCTTTGAGACC	95
	RP	GATGTCCTGCTCCTTGATGC	
Oncostatin M	FP	GTGGCTGCTCCAACTCTTCC	81
	RP	AGAGTGATTCTGTGTTCCCCGT	
Irisin	FP	GAGCCCAATAACAACAAGG	242
	RP	GAGGATAATAAGCCCGATG	
Actb	FP	CACTGTCGAGTCGCGTCC	102
	RP	CGCAGCGATATCGTCATCCA	

**Table 2 microorganisms-13-01009-t002:** List of bacterial species that showed significant differences between the five groups in feces.

Phylum	Class	Order	Family	*p*-Value	Effect Size
Firmicutes	Clostridia	Peptostreptococcales-Tissierellales	Peptostreptococcaceae	3.22 × 10^−11^	0.784391462
Firmicutes	Bacilli	Lactobacillales	Lactobacillaceae	3.94 × 10^−9^	0.71478921
Firmicutes	Bacilli	Lactobacillales	Streptococcaceae	1.28 × 10^−8^	0.694533816
Bacteroidota	Bacteroidia	Bacteroidales	Tannerellaceae	6.54 × 10^−7^	0.615028199
Bacteroidota	Bacteroidia	Bacteroidales	Rikenellaceae	1.28 × 10^−6^	0.599399448
Firmicutes	Clostridia	Oscillospirales	Ruminococcaceae	1.58 × 10^−6^	0.594330797
Firmicutes	Clostridia	Peptococcales	Peptococcaceae	4.08 × 10^−6^	0.570917473
Firmicutes	Clostridia	Christensenellales	Christensenellaceae	1.67 × 10^−5^	0.533237021

**Table 3 microorganisms-13-01009-t003:** List of bacterial species that showed significant differences between the five groups in MAM.

Phylum	Class	Order	Family	*p*-Value	Effect Size
Firmicutes	Bacilli	Lactobacillales	Streptococcaceae	1.63 × 10^−10^	0.753614
Bacteroidota	Bacteroidia	Bacteroidales	Rikenellaceae	6.18 × 10^−9^	0.697266
Firmicutes	Clostridia	Oscillospirales	Oscillospiraceae	5.30 × 10^−8^	0.657857
Firmicutes	Bacilli	Lactobacillales	Lactobacillaceae	1.94 × 10^−7^	0.631545
Firmicutes	Clostridia	Oscillospirales	Ruminococcaceae	4.47 × 10^−7^	0.613466
Firmicutes	Clostridia	Lachnospirales	Lachnospiraceae	6.71 × 10^−7^	0.60433

## Data Availability

The dataset is available upon request from the authors.

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
