# Peer review of "Fecal Microbiota Transplantation Using Donor Stool Obtained from Exercised Mice Suppresses Colonic Tumor Development Induced by Azoxymethane in High-Fat Diet-Induced Obese Mice"

_microorganisms, 2025, doi:10.3390/microorganisms13051009_

Round 1
Reviewer 1 Report
Comments and Suggestions for Authors
- Title: Change "obesity" to 'obese' in the title to make it 'high fat diet-induced obese mice'.
- Abstract: The role gut microbiota plays in the pathogenesis of cancers including colon cancer is no longer a "speculation". Replace "speculated" with a fitting word. The study subjects were divided into high-fat diet (HFD), normal diet (ND), normal diet plus exercise (ND+Ex), fecal microbial transplant from normal diet donor (FMT ND) and FMT from ND+Ex). The subsequent mention of HFD+FMT (ND) and HFD +FMT (ND+Ex) in the results is confusing. Need to use the same definitions of the groups throughout. The abstract can be shortened by limiting results to key findings.
- Keywords:
- Introduction: Although colorectal cancer is the leading cause of cancer deaths, it is not the most deadly (aggressive) cancer. Tumor is a general term. A tumor may be neoplastic or non-neoplastic and neoplastic tumor benign or malignant. Using cancer i.e. colorectal cancer would be preferably. Therefore, "tumor" should be changed to 'cancer' throughout. "micro-derived metabolites" in Line 48 is vague. It should be microbiota-, microbial- or microbiome-derived". Correct "colorectal and adenocarcinomas" to 'colorectal adenocarcinoma' Line 57. That intestinal microbiota is not only found in the lumen but also in the mucus is not a contradiction. Consider leaving out "Contrarily" in Line 60.
- Materials and methods: Replace "tumors" with cancer cells in line 109 to make it 'Colorectal cancer cells'. Also specify the cancer cell lines that were used i.e. where they were sourced from should be included in the manuscript. It is not sufficient to cite reference(s) 12-14. Abbreviation ND is supposed to stand for normal fat diet but is subsequently used for high-fat diet with FMT donor ND [HFD+FMT(ND)], which will confuse the readers. Similarly, [HFD+FMT(ND+Ex). Need to clarify the two HFD groups that received FMT from either ND alone or ND+Ex. Were there five groups from the start? Did the study have three groups on HFD from which two had FMT from either ND or ND+ER? Replace "quickly opened" in Line 152 with a specific timeline e.g. within x minutes or hours. Line 112-113 refers to autopsy and specimen collection and preparation at 26 weeks but Line 177-178 and Line 191-192 refer to collection of fecal samples at 28 weeks, and the mice were presumably still alive. It should be specified if the mice were euthanized at 26 weeks and specimens collected stored for 2 weeks. Also must clarify "Exercise started at 6 weeks of age for 2 weeks and continued for 20 weeks from 8 to 26 weeks". That "The diet was changed after acclimatization for 1 week" - did the five groups remain the same? Total number of weeks from 8 to 26 weeks should be 28 weeks. In Line 207 "category" should change 'categorical'. Remove "approximately" in Line 131 and leave it as 100mg, even if the amount was not exact.
- Aim: No concerns.
- Statistical analysis: That "the category data were analyzed using ch-square test" needs revision.
- Results: The HFD mice in Figure 2 is not just one group. Similarly, there are two groups of ND+Ex. Lines 219-221 introduce new groups FMT(ND) and FMT (ND+Ex), which will confuse the readers. Please correct and use same definitions throughout the manuscript.
- Illustrations: No concerns except Figure 1. Did all the 5 groups start at the same time?
- Discussion: Use 'cancer' instead of just "tumor".
- Acknowledgements: Please check and enter information.
- General: Need to review and revise some English grammar issues.
Author Response
Comments and Suggestions for Authors
- Title: Change "obesity" to 'obese' in the title to make it 'high fat diet-induced obese mice'.
Thank you very much for your advice. We rewrite the title.
- Abstract: The role gut microbiota plays in the pathogenesis of cancers including colon cancer is no longer a "speculation". Replace "speculated" with a fitting word.
Thank you very much for your advice. We rewrite the sentence in line 10.
The study subjects were divided into high-fat diet (HFD), normal diet (ND), normal diet plus exercise (ND+Ex), fecal microbial transplant from normal diet donor (FMT ND) and FMT from ND+Ex). The subsequent mention of HFD+FMT (ND) and HFD +FMT (ND+Ex) in the results is confusing. Need to use the same definitions of the groups throughout. The abstract can be shortened by limiting results to key findings.
Thank you very much for your suggestion. We rewrite HFD+FMT(ND) →HFD+FMT(ND alone).
- Keywords:
- Introduction: Although colorectal cancer is the leading cause of cancer deaths, it is not the most deadly (aggressive) cancer. Tumor is a general term. A tumor may be neoplastic or non-neoplastic and neoplastic tumor benign or malignant. Using cancer i.e. colorectal cancer would be preferably. Therefore, "tumor" should be changed to 'cancer' throughout.
Thank you very much for your suggestion. We rewrite the word in line 48.
"micro-derived metabolites" in Line 48 is vague. It should be microbiota-, microbial- or microbiome-derived".
Thank you very much for your suggestion. We rewrite the word “microbiome-derived” in line 49.
Correct "colorectal and adenocarcinomas" to 'colorectal adenocarcinoma' Line 57.
Thank you very much for your suggestion. We rewrite the word in line 59.
That intestinal microbiota is not only found in the lumen but also in the mucus is not a contradiction. Consider leaving out "Contrarily" in Line 60.
Thank you very much for your suggestion. We omit the word in line 61.
- Materials and methods: Replace "tumors" with cancer cells in line 109 to make it 'Colorectal cancer cells'. Also specify the cancer cell lines that were used i.e. where they were sourced from should be included in the manuscript. It is not sufficient to cite reference(s) 12-14.
Thank you very much for your suggestion. In this study, we did not use cancer cell lines, azoxymethane colon cancer model.
Abbreviation ND is supposed to stand for normal fat diet but is subsequently used for high-fat diet with FMT donor ND [HFD+FMT(ND)], which will confuse the readers. Similarly, [HFD+FMT(ND+Ex). Need to clarify the two HFD groups that received FMT from either ND alone or ND+Ex.
Thank you for your advice. We rewrite HFD+FMT(ND) → HFD+FMT(ND alone)
Were there five groups from the start?
Yes
Did the study have three groups on HFD from which two had FMT from either ND or ND+ER?
Yes it did.
Replace "quickly opened" in Line 152 with a specific timeline e.g. within x minutes or hours.
Thank you for your advice. We rewrite the words in line 158.
Line 112-113 refers to autopsy and specimen collection and preparation at 26 weeks but Line 177-178 and Line 191-192 refer to collection of fecal samples at 28 weeks, and the mice were presumably still alive. It should be specified if the mice were euthanized at 26 weeks and specimens collected stored for 2 weeks.
Thank you for your advice. 28 weeks was mistake. We rewrite to 26 weeks.
Also must clarify "Exercise started at 6 weeks of age for 2 weeks and continued for 20 weeks from 8 to 26 weeks".
Thank you for your suggestion. We rewrite the sentence.
That "The diet was changed after acclimatization for 1 week" - did the five groups remain the same? Total number of weeks from 8 to 26 weeks should be 28 weeks.
Thank you for your advice.
In Line 207 "category" should change 'categorical'.
Thank you for your advice. In line 212.
Remove "approximately" in Line 131 and leave it as 100mg, even if the amount was not exact.
Thank you for your advice. We omit “approximately” in line 141.
- Aim: No concerns.
- Statistical analysis: That "the category data were analyzed using chi-square test" needs revision.
Thank you very much for your advice. We rewrite the sentence in line 214.
- Results: The HFD mice in Figure 2 is not just one group. Similarly, there are two groups of ND+Ex. Lines 219-221 introduce new groups FMT(ND) and FMT (ND+Ex), which will confuse the readers. Please correct and use same definitions throughout the manuscript.
Thank you very much for your advice. We rewrite the group name.
- Illustrations: No concerns except Figure 1. Did all the 5 groups start at the same time?
Thank you very much for your suggestion. All the 5 groups start at the same time.
- Discussion: Use 'cancer' instead of just "tumor".
Thank you for your advice. We rewrite the word.
- Acknowledgements: Please check and enter information.
Thank you very much for your suggestion.
- General: Need to review and revise some English grammar issues.
Thank you very much for your advice.

Reviewer 2 Report
Comments and Suggestions for Authors
The authors present some work on the the effect of fecal transplantation on "tumor" development in a mouse model of colon cancer. There are a number of major concerns with the manuscript.
major concerns:
- The authors have used the number of polyps as an index of cancer development. There is no histology of these polyps making it impossible to evaluate their tumor development.
- I do not understand Tables 2 and 3. Its most unclear as to which groups are being compared.
- For table 2 "erysipelstrichaceae" is mentioned in the text associated with table 2 but it is not listed in Table 2.
- What are the units (Y-axis) for all parts of Figure 7?
- There are significant differences in lactic and butyric acid for normal diet mice versus FMT mice but the authors do not discuss.
- Lines 358 - 363. In the present study, we confirm that fecal transplantation increases the number of 358
Lactobacillaceae, which promotes the production of the antioxidant glutathione and anti-
angiogenic factors, and, through the downregulation of polyamine components, it in
creases inflammation, DNA damage, and tumor load levels [37] and further delays tumor
development by increasing the NK and CD8+ T cell infiltration and increasing IFNγ pro
duction in the cancer microenvironment [38]. This sentence doe snot make much sense as you say that Lactobacillaceae promote DNA damage and also delays tumor development.
Minor concern:
I assume you mean Succinic acid not succunic acid (Figure 6).
Comments on the Quality of English LanguageThere are many typographical errors in the manuscript.
Author Response
Comments and Suggestions for Authors
The authors present some work on the the effect of fecal transplantation on "tumor" development in a mouse model of colon cancer. There are a number of major concerns with the manuscript.
major concerns:
- The authors have used the number of polyps as an index of cancer development. There is no histology of these polyps making it impossible to evaluate their tumor development.
Thank you for your comments. We add the new sentence about histological analysis, picture in additional figures, and reference in line 118.
- I do not understand Tables 2 and 3. Its most unclear as to which groups are being compared.
Thank you for your comments. We rewrite the text and table legends.
- For table 2 "erysipelstrichaceae" is mentioned in the text associated with table 2 but it is not listed in Table 2.
Thank you very much for your advice. The word “erysioelstrichaceae” is mistake, so we rewrite the sentence. About “erysipelstrichaceae” we add the figure legends of figure 8.
- What are the units (Y-axis) for all parts of Figure 7?
Thank you for your advice. Y-axis showed the expression of cytokines when the expression of β-actin is set to 1 as an intrinsic control.
- There are significant differences in lactic and butyric acid for normal diet mice versus FMT mice but the authors do not discuss.
Thank you for your comments.
About lactic acid, there are not significant between ND vs. FMT mices. On the other hands, about butyric acid, there are significant difference between ND vs. FMT mice, similarly succunic acid, acetic acid, and propionic acid. Thus, we add the new sentence about them in both results and discussion sections.
- Lines 358 - 363.
In the present study, we confirm that fecal transplantation increases the number of 358.
Lactobacillaceae, which promotes the production of the antioxidant glutathione and anti-
angiogenic factors, and, through the downregulation of polyamine components, it in
creases inflammation, DNA damage, and tumor load levels [37] and further delays tumor
development by increasing the NK and CD8+ T cell infiltration and increasing IFNγ pro
duction in the cancer microenvironment [38]. This sentence does not make much sense as you say that Lactobacillaceae promote DNA damage and also delays tumor development.
Thank you for your comment. We omit the sentence and reference [37], and rewrite the sentence.
Minor concern:
I assume you mean Succinic acid not succunic acid (Figure 6).

Round 2
Reviewer 2 Report
Comments and Suggestions for Authors
The authors have for the most part answered my questions. there are still a couple of points.
- Figure 5(A) it is still labeled succunic acid. There is no label on the Y-axis.
- Although the authors have provided a histopathological slide in supplementary material, the findings are not described in the results section.
Author Response
Comments and Suggestions for Authors
The authors have for the most part answered my questions. there are still a couple of points.
Thank you for your kind comments.
- Figure 5(A) it is still labeled succunic acid. There is no label on the Y-axis.
→Thank you for your comment. I am sorry our mistake. I add new figure5A.
- Although the authors have provided a histopathological slide in supplementary material, the findings are not described in the results section.
→Thank you for you advice. I add the new sentence about it in line 233-235.
